# Polyimide Films Based on β-Cyclodextrin Polyrotaxane with Low Dielectric and Excellent Comprehensive Performance

**DOI:** 10.3390/polym16070901

**Published:** 2024-03-25

**Authors:** Xuexin Zhang, Yao Dou, Liqun Liu, Meixuan Song, Zhenhao Xi, Yisheng Xu, Weihua Shen, Jie Wang

**Affiliations:** 1State Key Laboratory of Chemical Engineering, East China University of Science and Technology, Shanghai 200237, China; xuexinzhang2023@163.com (X.Z.); y82200090@mail.ecust.edu.cn (Y.D.); leoliu0811@163.com (L.L.); 15854768116@163.com (M.S.); whshen@ecust.edu.cn (W.S.); 2Shanghai Key Laboratory of Multiphase Materials Chemical Engineering, East China University of Science and Technology, Shanghai 200237, China

**Keywords:** colorless polyimide, beta-cyclodextrin, semi-aromatic, dielectric constant

## Abstract

In order to prepare polyimide (PI) films with a low dielectric constant and excellent comprehensive performance, a two-step method was employed in this study to integrate β-cyclodextrin into a semi-aromatic fluorine-containing polyimide ternary system. By introducing trifluoromethyl groups to reduce the dielectric constant, the dielectric constant was further reduced to 2.55 at 10 MHz. Simultaneously, the film exhibited noteworthy thermal stability (a glass transition temperature exceeding 300 °C) and a high coefficient of thermal expansion. The material also demonstrated outstanding mechanical properties, boasting a strength of 122 MPa and a modulus of 2.2 GPa, along with high optical transparency (transmittance reaching up to 89% at 450 nm). Moreover, the inherent high transparency of colorless polyimide (CPI) combined with good stretchability contributed to the attainment of a low dielectric constant. This strategic approach not only opens up new opportunities for novel electroactive polymers but also holds potential applications in flexible displays, circuit printing, and chip packaging.

## 1. Introduction

In recent years, the escalating advancements in 5G technology and integrated circuitry have precipitated a surge in the prominence of polyimide films. This heightened attention is primarily attributed to the extraordinary performance characteristics inherent in polyimides, encompassing elevated thermal stability, robust mechanical strength, chemical inertness, and superior dielectric properties [1]. This multifaceted material has garnered substantial interest for its versatile utility in a spectrum of high-value applications, spanning film, fiber, foam, composite materials, and adhesives. Noteworthy applications encompass tissue scaffolds [2], gas separation membranes [3], and battery membranes [4], positioning polyimide as a material of paramount significance within the realms of advanced materials. This surge in interest underscores its pivotal role in addressing the evolving demands of contemporary technological landscapes, particularly in domains such as 5G infrastructure and integrated circuitry. Consequently, this nuanced exploration positions polyimide as a material with profound implications for diverse scientific and engineering applications.

Dielectric materials assume a pivotal role in propelling the evolution of wireless technology, particularly as the landscape of the Internet of Things (IoT) undergoes exponential growth. The escalating demand for wireless connectivity in electronic devices, exemplified by 5G communication, intelligent transportation systems, and remote healthcare, underscores the criticality of dielectric materials in advancing these domains [5]. The global wireless data transmission market is experiencing a remarkable surge, necessitating the adoption of millimeter-wave (mW) frequencies spanning from 30 GHz to 300 GHz, and potentially higher, to accommodate this unprecedented growth [6]. The diminishing dimensions of Ultra Large-Scale Integration (ULSI) exacerbate interconnect delay time, which is primarily attributable to the parasitic capacitance of the interconnect. Mitigating this challenge mandates the implementation of low-κ dielectrics to attenuate the parasitic capacitance of the interconnection [7]. Nonetheless, the constraints inherent in millimeter-wave communication, such as limited transmission distance, severe signal attenuation, and susceptibility to obstruction, pose formidable challenges to the effective deployment of wireless data transmission in these frequency bands. Addressing these challenges necessitates a rigorous analysis of signal delay time, a parameter calculable through a defined equation [8,9,10]. This scholarly exploration seeks to elucidate the intricate interplay between dielectric materials, transmission frequencies, and the associated challenges encountered in the pursuit of advancing wireless communication technologies. In doing so, it contributes to the overarching discourse within the domain of advanced materials.
(1)$${t=\mathrm{RC}=2\rho \varepsilon \varepsilon }_{0}\left[\frac{{4L}^{2}}{P^{2}}+\frac{L^{2}}{T^{2}}\right]$$

Within the domain of electronic materials, the pursuit of components conducive to high-speed and low-delay signal transmission stands as a paramount endeavor. This holds particularly true in the sphere of high-frequency applications, where the dielectric medium assumes a pivotal role. Traditional silica-based materials, although prevalent, are encumbered by inherent limitations arising from their dielectric constant (Dk) and dielectric loss factor (Df) properties [11,12]. In light of these constraints, polyimide has emerged as a preeminent choice within the realm of high-performance insulation materials. Its extensive utilization spans diverse applications, including but not limited to flexible displays [9], circuit printing [10], and chip packaging [11]. The adoption of polyimide in these contexts is driven by its capacity to address the shortcomings associated with traditional silica-based materials. This deliberate selection is informed by polyimide’s advantageous dielectric characteristics, contributing to enhanced performance and reliability in high-frequency electronic applications.

Nevertheless, when PI is employed in domains like circuit-printing films and semiconductor coatings, precise control or enhancement of certain intrinsic PI properties becomes imperative. Specifically, meticulous regulation of the dielectric constant is essential. Traditional polyimides typically exhibit a dielectric constant within the range of 3.4–3.5. However, stringent requirements necessitate lower values (<3) to mitigate issues associated with Ultra Large-Scale Integration (ULSI). Additionally, the limited hydrophobicity of traditional polyimides constrains their applicability in electrical materials. The molecular architecture of the polyimide polymer emerges as a decisive factor influencing its dielectric properties. Notably, the impact of polar functional groups and non-polar functional groups on dielectric characteristics is dichotomous. Achieving optimal dielectric properties often mandates the incorporation of non-polar functional groups into the polymer chain. This strategic modification serves as a means to fine-tune and enhance the dielectric performance of polyimide, aligning it with the exacting requirements of advanced applications in circuit printing and semiconductor coatings.

In the pursuit of optimizing the dielectric properties of polyimide (PI) for high-speed signal transmission, a fundamental imperative lies in the precise molecular engineering of the polymer. This intricate process involves the deliberate introduction of non-polar functional groups into the polymer chain, strategically aimed at achieving the requisite reduction in the dielectric constant. This meticulous molecular engineering approach not only serves to ameliorate the material’s dielectric performance but also concurrently augments its hydrophobic characteristics, thereby bestowing a dual advantage for electronic applications. The strategic incorporation of non-polar functional groups at the molecular level is undertaken with a specific focus on tailoring the dielectric properties of the polyimide. By modulating the molecular structure in this manner, a targeted reduction in the dielectric constant is achieved, aligning the material with the stringent requirements of high-speed signal transmission. Simultaneously, this tailored molecular engineering imparts enhanced hydrophobicity to the polyimide, fortifying its resilience in diverse electronic applications.

The strategies employed for reducing permittivity encompass two primary methodologies: the introduction of fillers characterized by low permittivity and the augmentation of molecular free volume within polymers. Notably, the introduction of fluorine groups into diamine or dianhydride monomers represents a well-established approach in the synthesis of fluorine-containing polyimide. The robust electronegativity exhibited by fluorine atoms contributes significantly to diminishing molar polarizability and intermolecular forces. Furthermore, the strategic incorporation of fluorine groups into the side chains of the polyimide structure serves to effectively diminish the stacking density of molecular chains. This deliberate structural modification imparts an increased free volume to the system, thereby expanding the molecular interstices and leading to a reduction in the dielectric constant through multiple synergistic effects. The introduction of fluorine elements and the concurrent elevation of the Fractional Free Volume (FFV) represent classical yet effective methodologies for permittivity reduction in polyimide.

Two strategies to reduce the dielectric constant of polyimide (PI) based on the Clausius-Mossotti Equation (2) involve diminishing molecular polarizability or dipole density. Zhang [12] explored the augmentation of free volume by leveraging the secondary relaxation behavior exhibited by polymer chains. However, certain drawbacks are associated with conventional hole-making methods. For instance, caustic soda etching poses high operational risks and creates environmental pollution. Furthermore, the incorporation of polyethylene glycol (PEG) may decrease the polymer density but is susceptible to collapse during thermal degradation [13], leading to compromised mechanical properties. Notably, as pores contract, the dielectric constant of the PI film tends to increase. Efforts to address these limitations include the utilization of ionic liquids as porogens, which ameliorate process drawbacks [14]. However, challenges persist as small molecules embedded in the polymer may not be entirely removed, impacting dielectric properties. Zhuang introduced a novel and controllable selective etching strategy to fabricate yolk–multishell mesoporous silica nanoparticles, albeit involving HF etching of SiO_2_ layers [15]. In addition, the integration of Cage polyhedral oligomeric silsesquioxane (POSS) hollow nanoparticles into polymers has been explored to develop low-k nano hybrid materials. Nevertheless, this approach is constrained by limitations such as small particle size, high costs, and synthetic challenges associated with functionalized POSS monomers [16,17].
(2)$$\mathrm{Dk}=\frac{\left[1+2\left(\frac{P}{V}\right)\right]}{\left[1-\left(\frac{P}{V}\right)\right]}$$
where P (cm^3^ mol^−1^) and V (cm^3^ mol^−1^) are the molar polarization of polymer functional groups and the molar volume of polymer functional groups (free volume).

Based on these two strategies, a pseudopolyimiderotaxanes structure was orchestrated through the incorporation of an aliphatic ring and a trifluoromethyl group into the polyimide chain, coupled with the encapsulation of β-cyclodextrin [18]. Cyclodextrin (CD), an oligosaccharide with cyclic characteristics, renowned for its exceptional solubility and pore structure, has been established as capable of forming rotaxane structures. The introduction of cyclodextrin into the polymer backbone instigates an augmentation in the intermolecular spacing, concurrently diminishing molecular polarizability. Leveraging 4,4′-hexafluoroisopropylidenediphthalic anhydride (H″PMDA), the pseudopolyrotaxane structure involving β-cyclodextrin serves to reduce dipole density, thereby effecting a decline in the dielectric constant of polyimide films from 2.8 to 2.65. This concerted effect results from the coordination of two distinct mechanisms: the widening of intermolecular gaps induced by cyclodextrin and the reduction in dipole density facilitated by the pseudopolyrotaxane structure. Consequently, the dielectric breakdown strength of the polymer experiences enhancement upon doping. It is noteworthy, however, that, to the best of our knowledge, the impact of the pseudopolyrotaxane structure on the dielectric constant of polyimide films has not been systematically investigated. To address this knowledge gap, our study delves into the influence of the pseudopolyrotaxane structure on the dielectric constant of polyimide films. Through meticulous adjustments to the pseudopolyrotaxane structure within the polyimide polymer main chain, we successfully engineered low-dielectric polyimide films featuring the pseudopolyrotaxane structure.

## 2. Materials and Methods

### 2.1. Materials

The compound 1R,2S,4S,5R-Cyclohexanetetracarboxylic Dianhydride (99.8%) was purchased from Weihai Sun won Kosun new materials (Weihai, China). 4,4′-diaminodiphenyl ether (99.78%) was purchased from Shanghai bidde medical (Shanghai, China). 2,2-bis [4-(4-aminophenoxy) phenyl] hexafluoro propane (99.99%) was purchased from Shanghai bidde medical (Shanghai, China). β-Cyclodextrin (98%) was purchased from Shanghai McLin biotech (Shanghai, China). and pyridine (99.5%) was purchased from Shanghai Merrill Biotech (Shanghai, China). Acetic anhydride (98%), *N*,*N*-dimethylacetamide (98%), Anhydrous ethanol, Deionized water, β-cyclodextrin (β-CD), and 1R,2S,4S,5R-Cyclohexanetetracarboxylic Dianhydride (H″PMDA) were treated in a vacuum oven at 140 °C for 10 h, β-cyclodextrin (β-CD), 4,4′-diaminodiphenyl ether (ODA) and 2,2-bis[4-(4-aminophenoxy)phenyl]hexafluoropropane (HFBAPP) was treated in a vacuum oven at 60 °C for 6 h, and solvent *N*,*N*-dimethylacetamide (DMAc) was treated with 4A molecular sieve for 24 h.

### 2.2. Preparation the Complexation of Diamine Monomer with β-Cyclodextrin

The steps for preparing the nested cyclodextrin diamines are shown in Figure 1. Two kinds of diamine (ODA, HFBAPP) monomers with nested cyclodextrin structures were synthesized.

The preparation process included adding 0.5016 g ODA (2.5 mmol) to a 100 mL dry three-necked flask containing 20 mL DMAC, then stirring it in a 5 °C water bath under the protection of nitrogen gas until ODA was completely dissolved. Then, 1.2987 g HFBAPP (2.5 mmol) was added to the solution; the mixture was then kept in a 5 °C water bath under nitrogen protection and stirred again until HFBAPP was completely dissolved. After that, β-CD inclusion complex diamine solution was obtained by adding 5.7613 g β-CD for 24 h. The distinct component ratios were employed in the preparation process, as delineated in Table 1 (PI-0, PI-3, PI-5, PI-7, PI-10, and inclusion complex polyimide with varying β-CD content. Appendix A: Synthesis of polyimide with different degrees of addition (β-CD)).

### 2.3. Synthesis of Soluble β-Cyclodextrin-Polyimide Composite Films

A two-step synthesis pathway was employed for the fabrication of polyimide (PI), specifically the inclusion complex polyimide. The process entailed condensation polymerization from PAA followed by temperature-programmed heating, as elucidated in Figure 2, illustrating the corresponding reaction mechanism. The method comprised the following steps: introducing the nested diamine monomer into a 100 mL dry tri-neck flask containing 20 mL DMAC, then stirring in a 5 °C water bath under the protection of nitrogen for 24 h until β-CD nested diamine completely dissolved. Subsequently, 2.3333 g H″PMDA (0.0102 mol) was added and stirring continued in a 5 °C water bath under the protection of nitrogen for an additional 24 h, culminating in the formation of a transparent and viscous PAA solution.

After the reaction finished, the PI solution was allowed to stand for 45 min to facilitate the thorough elimination of microporous bubbles within the PAA. This precautionary measure aimed to prevent potential bubbling defects that may arise during the subsequent drying process. Subsequently, the PAA solution was carefully spun onto a pristine glass plate, ensuring the complete dryness of the plate and its strict horizontal alignment to mitigate variations in the resulting PI film thickness. In order to obtain a flexible PI film without inducing undesirable coloration, sequential stages were performed at 80 °C for 3 h, followed by incremental temperature steps of 120 °C, 150 °C, 180 °C, 200 °C, with each temperature level maintained for 1 h during the drying process.

### 2.4. Preparation of β-Cyclodextrin PI Composite Films

The films designated for thermal analyses were prepared by depositing polyimide DMAC solutions containing polymer (20 wt%) onto a glass substrate. Subsequently, the films were subjected to a drying process at 80 °C for a duration of 3 h. Following this initial drying stage, the temperature was incrementally raised to 120 °C over a 2 h period, followed by additional drying periods at 120 °C, 150 °C, and 200 °C, with each designated temperature sustained for 1 h.

### 2.5. Characterization of β-Cyclodextrin PI Composite Flims

^1^HNMR spectra (400 MHz) were recorded on AVANCE III 400 instrument from Bruker (Fällanden, Switzerland) in dimethyl sulfoxide-*d_6_* at 25 °C. FT-IR measured by Nicolet FTIR 6700 from Thermo Fisher (Lenexa, KS, USA), with scanning wavenumbers ranging from 4000 to 400 cm^−1^. These findings were used to detect the characteristic absorption peaks of PI films.

Inherent viscosities were measured with a SI Analytics DIN viscometer (Mainz, Germany) in poly(amic acid) DMAc solution and in polyimide DMAc solution at a concentration of 0.5 g/dL at 30 °C.

Gel permeation chromatography (GPC) was carried out using a PL-GPC50 from Agilent (Colorado Springs, CO, USA); column: 2 × Agilent PLgel, 5 µm, MIXED-C 300 mm × 7.5 mm (part number PL1110-6500), operating from ambient to 40 °C, concentration 0.1 wt./vol%, and *N*,*N*-Dimethylformamide (DMF) as an eluent detector. Molecular weight is calibrated against the standard molecular weight of polystyrene.

Wide-angle X-ray diffraction (WAXS) was measured with 18 KW/D/max2550 VB/PC from Rigaku Electric (Tokyo, Japan) with Copper Target (radiation wavelength: 1.54 Å, temperature of data collection: 25 °C, collection range 2θ: 10–70°) to analyze the internal aggregation structure of films.

The microstructures of films were examined using a scanning electron microscope (Nova Nano SEM 450, Thermo Fisher, Lenexa, KS, USA). Resolution: High Vacuum: 1.0 nm@15 KV; 1.4 nm@1 KV. Electron microscope acceleration voltage: 200 V–30 KV.

Thermal gravimetric analysis (TGA) was measured using a PerkinElmer TGA 8000 (Shelton, CT, USA) to test the thermostability, conducted at a heating rate of 10 °C min^−1^ and a nitrogen flow rate of 100 mL min^−1^. The glass transition temperature (DSC) was obtained by a PerkinElmer DSC 8500 (Shelton, CT, USA) in a temperature range of 25–450 °C under nitrogen gas, and the heating rate was 10 °C/min.

The mechanical properties of the PI composite films were evaluated using a TSKL-5T dual column tensile testing machine from Tinius kuli (Suzhou, China) at a speed of 50 mm/min. The tensile modulus (E), tensile strength (rb), and fracture elongation (eb) of the PI specimens (film dimensions, 30 mm length, 3 mm width, typically 20 μm thickness) were investigated. In this case, the specimens were taken from a large sheet (10 × 10 cm^2^) without any defects such as small bubbles. The specific stretching requirement was 10 mm/min at 25 °C.

UV/visible spectrometer measurements were acquired using a PerkinElmer Lambda 950 UV/Vis spectrometer (Shelton, CT, USA). The solubility behaviors of polyimides in various kinds of solvents were investigated by dissolving 10 mg of powdery polymer samples in 1 mL of solvent, either at room temperature or at elevated temperatures.

Dielectric properties were tested by a Concept 80 Novocontrol GmbH (Montabaur, Germany) impedance analyzer at various frequencies ranging from 10^2^ Hz to 10^6^ Hz.

The dielectric properties were measured at 25 °C using a precision impedance analyzer (Cocept 80) in the frequency range of 40–10 MHz with PI films cut into circular sheets with radii of 10 mm and coated with gold double-side-sprayed. The relative dielectric permittivity (**ε**) of the films was calculated from Equation (1) [20].
(3)$$\varepsilon =\frac{\mathrm{Cd}}{{A\varepsilon }_{0}}$$
where ε is the dielectric permittivity of the PIs, ε_0_ is the vacuum dielectric permittivity (8.854 × 10^−12^ F/m), and d and A are the film thickness and electrode area, respectively.

## 3. Results

### 3.1. The Complexation of Diamine Monomer with β-Cyclodextrin

The solution of β-CD (5.7613 g) dissolved in DMAC at 5 °C was employed, and ODA (0.5016 g, 2.5 mmol) and HFBAPP (1.2987 g, 2.5 mol) was added portionwise under constant stirring. The reaction mixture was refluxed for 6 h and was vigorously agitated using a mechanical stirrer. The β-CD/ODA components transitioned into a homogeneous solution, which was further stirred for an additional 24 h. The other researchers synthesized alicyclic polyimides as shown in Table 2.

Table 3 presented the outcomes of pseudopolyimiderotaxanes. The weight-average molecular weights of inclusion polyimides (Appendix A) were confined to a range of 1.9 × 10^5^~4.0 × 10^5^. The diminished yield and molecular weight observed for PI-CD2 were attributed to the reduced solubility of ODA at room temperature in DMAC. Notwithstanding this challenge, the film formation process for all included polyimides on glass substrates exhibited commendable performance. The resulting films demonstrated transparency and exhibited optimal conditions, underscoring the robustness of the film formation procedure employed in this study.

The preparation of pseudopolyimiderotaxanes entails a straightforward procedure involving the reaction of a diamine solution containing nested β-cyclodextrin with a dianhydride solution through mechanical stirring at low temperatures, as illustrated in Figure 1. In comparison to pore-forming techniques, such as the utilization of etched silica nanoparticles and the pyrolysis of poly(vinyl alcohol), this method eliminates the need for highly corrosive solvents like hydrofluoric acid and allows for the decomposition of β-cyclodextrin at lower temperatures. To elaborate further, the incorporation of cyclodextrins serves to augment the free volume within the polymer backbone. Pore-forming, achieved by elevating temperatures up to 240–260 °C represents a milder, safer, and more efficient alternative. The avoidance of harsh solvents and the reduced decomposition temperature of β-cyclodextrin render this approach advantageous over traditional methods. Figure 1 illustrates the mechanism underlying the high transmittance of polyimide films, attributed to the cooperative action of charge transfer complexes between the attenuated main chains. Initially, the amino group (-NH_2_) in (HFBAPP) reacts with the anhydride group (-C=O) in (H″PMDA), forming a polyamido acid (PAA) (-CH-N-) precursor at low temperatures. Subsequently, polyimide is obtained through pyridine/acetic anhydride-catalyzed closed-loop dehydration. Furthermore, the enhancement of mechanical properties is attributed to hydrogen bonding interactions involving phenolic hydroxyl groups on nested β-CD and fluorine atoms within the polyimide molecule. α-Cyclodextrin’s inner diameter is 0.5 nm, while γ-Cyclodextrin’s inner diameter is 0.8 nm. The chain spacing of the polyimide film was calculated from the angular values of the diffraction peaks of the X-ray diffraction spectra as 5.1013–5.2688 Å so that α-cyclodextrin could not form a nested structure with it and γ-cyclodextrin would detach from it during subsequent washing due to the large diameter of the inner cavity.

### 3.2. Synthesis and Characterization of β-Cyclodextrin Polyimide Composite Films

As shown in Figure 1, the inclusion compound of ODA/HFBAPP and β-CD was obtained by dissolving ODA and HFBAPP in DMAC and kept in a water bath of 5 °C under nitrogen protection, then stirred until complete dissolution of HFBAPP/ODA was achieved. All of the ICs exhibited ready solubility in low-boiling-point polar solvents such as chloroform (CHCl_3_) and dichloromethane (CH_2_Cl_2_). All the compounds of β-CD-PI composites and β-CD were characterized by ^1^H NMR (Appendix A) and FTIR. In ^1^H NMR spectra (Figure 3) of β-CD, β-CD-H″PMDA-HFBAPP/ODA-PI and β-CD-H″PMDA-HFBAPP/ODA-PI compounds, compared with PI without β-CD, there were more peaks of β-CD between 3.5–6.0 ppm in pseudopolyimiderotaxanes system. Notably, most protons of ^1^H NMR were shifted upfield slightly: the signal of PI-CD1 (δ_3C-H_ = 4.37, δ_5C-H_ = 4.85) was shifted from the corresponding signals of β-CDs (δ_3C-H_ = 4.47, δ_5C-H_ = 4.87) [19]. It is interesting that H3 and H5 were oriented towards the interior of the cavity, and therefore, changes in its signal provide information on the inclusion process.

In order to further validate the conformational assignment depicted in Figure 2 as the structure of β-CD/PI, reference was made to the ^1^H NMR spectrum of β-CD (Appendix A). The singlet at δ = 5.90 ppm was attributed to H1, while the multiplets at δ = 4.47 ppm and δ = 4.87 ppm were ascribed to H3, H5, and H6. Furthermore, the singlet at δ = 3.62 ppm was designated for H2, and the triplet state at δ = 3.53 ppm was assigned to H4. The labelling Ha on the PI represents the protons of the aliphatic ring on H"PMDA at δ = 3.15. These assignments serve to reinforce the structural elucidation of the β-CD/polyimide composites.

Figure 3 shows the ^1^H NMR signals of H2 to H6 protons. Upon the recombination of β-CD/PI, discernible shifts in these signals were observed. Notably, there is a significant field displacement (~0.1 ppm) for H3 protons. The displacement of H5 protons towards the field is relatively small (~0.02 ppm), while the impact on H2 and H4 protons is minimal (<0.01 ppm). The positioning of H3, H5, and one of the H6 protons within the CD cavity is evident in Figure 3, and the upward movement of these protons suggests the encapsulation of certain fragments of PI molecules within the β-CD cavity. Since the protons of H″PMDA were not affected (Figure 4), we propose that only the aromatic protons of ODA were included in the cavity. Therefore, the ^1^H NMR experimental results indicate that there are aromatic protons of ODA in the PI system in the CD cavity, and the aromatic part is located in the cavity.

The NOESY spectrum of pseudopolyimiderotaxanes is shown in Figure 4. A noteworthy observation is the presence of cross peaks between the aromatic proton (δ = 7.37 ppm) and the inner protons of β-cyclodextrin (β-CD) (δ = 4.47 ppm), providing conclusive evidence of the inclusion of the aromatic moiety within the β-CD cavity. Interestingly, the protons of β-CD (notated as H3 and H5 in Figure 4), initially observed as a singlet at 4.47 ppm and 4.87 ppm, displayed cross peaks with the H3 and H5 protons of β-CD. This interaction induced 0.1 ppm upfield shifts for H3 and 0.02 ppm upfield shifts for H5 protons. While the H3 and H5 protons in native β-CD manifested as a multiplet, in β-CD/polyimide (PI), the H5 protons shifted upfield, and the H6 protons were relatively unaffected, resulting in the spatial separation of these peaks, as illustrated in Figure 4. The NOESY spectrum unequivocally demonstrates that the aromatic group is positioned within the CD cavity in β-CD/PI, providing robust support for conformation C.

The synthesis process of β-CD-polyimide(β-CD-PI) composite films was shown in Figure 2. The polymerization of β-CD, ODA, HFBAPP and H″PMDA was performed by chemical imidization. The PAA solution, consisting of pyridine and Ac_2_O, was amalgamated to yield a homogeneous solution. Following a 24 h chemical imidization process, the resultant product was precipitated in methanol and deionized water, followed by dissolution in DMAC after vacuum drying. Subsequently, the β-CD-PI composite films were fabricated through a casting and thermal processing method, followed by solvent removal. The composited films underwent a final step in which the doped β-CD was completely eliminated through washing with water. It is noteworthy that the increase in the volume fraction of the polymer is defined as the content of β-CD in the PI matrix.

The distinctive absorption bands corresponding to essential functional groups in polyimide, such as -CO, -C-N-C, and -C-F, were clearly identified at 1700 cm^−1^, 1360 cm^−1^, and 1230 cm^−1^ in the FTIR spectrum (Figure 5). These characteristic peaks serve as unequivocal indicators of the successful fabrication of PI films [26]. The internal aggregation structure of polymer molecules within the PI films was further elucidated through XRD analysis (Figure 6). A broad peak observed between 16.82° to 17.38° in both β-cyclodextrin (β-CD)-free polyimide and inclusion polyimide signifies the overall structural integrity of the polyimides. It is imperative to note that the chain spacing (d-spacing) derived from XRD data reflects the degree of aggregation between polymer segments. The molecular packing in pure PI films and β-CD-PI composite films underwent detailed analysis using the Bragg equation [27], and specific data were documented in Appendix A.

XRD showed the results of pseudopolyimiderotaxanes. The weight-average molecular weights of inclusion polyimides fell within the range of 200,000–400,000, which indicated that the chemical imidization method fits the system. The satisfied yield and high molecular weight of polyimide were due to the high solubility of β-CD/ODA/HFBAPP at 5 °C in DMAC. Moreover, all the composite films exhibited well-formed structures on glass substrate.

According to Table 3, it can be observed that the viscosity of the four PAA solutions reached the level characteristic of high-molecular-weight polymers. However, the incorporation of β-cyclodextrin significantly diminished the molecular weight of PI. The number-average molecular weight decreased from 2.9 × 10^5^ to approximately 1.5 × 10^5^, while the weight-average molecular weight declined from 4.0 × 10^5^ to around 2.1 × 10^5^. This phenomenon could be attributed to the subtly conical hollow cylindrical structure of cyclodextrin molecules, which intricately nestle onto the PI molecular chains, thereby forming a stable polyrotaxane structure. The trifluoromethyl group of the diamine HFBAPP and the acyl imide ring on the PI molecular chains possess a substantial spatial volume, acting as a “plug” that effectively hinders the dislodgment of β-CD from the PI molecular chains. Consequently, this “plug” formation contributes to the stability of the polyrotaxane structure. An additional factor to consider is the larger volume of beta-cyclodextrin, which potentially restricts the free movement of diamine and dianhydride monomers in the solution, reducing the probability of their collision.

Another possibility is that β-CD may not exclusively form a 1:1 inclusion complex with the diamine monomer; instead, multiple β-CD molecules may bind to a single diamine monomer, thereby shielding its functional groups and impeding their reactivity with the acid anhydride groups. This interference in molecular chain growth results in a diminishing trend in chain length with an increasing quantity of β-CD. Additionally, the polymer dispersity index (PDI) of PI-CD3 anomalously increases to 1.77, indicating a wide molecular weight distribution with a higher proportion of smaller molecules. This is likely caused by the two aforementioned factors leading to a reduction in molecular weight.

The cross-section of pristine polyimides without β-CD addition reveals a smooth and dense morphology devoid of any discernible holes or defects, as depicted in Figure 6. This observation suggests the attainment of polyimides with high molecular weight. As β-CD is progressively introduced, the cross-section of PI undergoes a transition from smooth to rough. The SEM images illustrate non-uniform cross-sections of PI composites, exhibiting cracks and a scaly appearance, which indicates the presence of a ductile fracture [28]. Moreover, the section roughness of PI composites exhibits a gradual increase corresponding to the escalating β-CD doping amounts ranging from 1.25 to 5 mmol, as illustrated in Figure 6.

This is mainly due to the enhanced reactivity of the chair configuration compared to the ship configuration [22], facilitating the formation of long-chain polymers. Furthermore, the aliphatic structure of cyclohexane tetracarboxylic dianhydride contributes to the weakening of the charge transfer between molecular chains, owing to its twisted conformation [29]. This structural feature increases the spatial separation between molecular chains. Additionally, the HFBAPP diamine monomer’s side chain, which incorporates a trifluoromethyl group, possesses a significant free volume [30]. This characteristic further contributes to increasing the distance between polymer molecular chains, hindering the facile sliding of β-CD under stress. Consequently, this impediment results in a larger polymer-chain free volume and the formation of irregular cross-sections.

The colorless polyimide was obtained through a condensation reaction involving the carboxyl of dianhydride and the amino group of diamine, with the polyamide acid completing the dehydration closed-loop.

The degree of cyclodextrin addition in PI products under various preparation conditions is shown in Table 1. The experiment yielded a dielectric constant as low as 2.65, significantly surpassing values reported in similar literature [31]. Throughout the reaction process, adjusting the feed ratio of β-CD and diamine allowed for the synthesis of β-CD/PI with varying β-CD content, enabling the production of films with diverse properties.

### 3.3. Mechanical Properties of β-Cyclodextrin PI Composite Films

For each structure, stress–strain curves were averaged with a strain range of 6.3–7.4%. All the β-CD/PI films exhibited a tensile strength range of 98–145 MPa, a tensile modulus ranging from 1.87–2.23 GPa, and an elongation at break ranges greater than 5%. In addition, the co-PI film with β-CD molar ratio of 20% showed the best tensile strength of 145 MPa.

The films containing β-CD possessed a significantly lower tensile strength, and even larger elongation at break, which was ascribed to its chair structure that resembled that of H″PMDA. Mechanical performance data for β-CD/polyimide (PI) films are presented in Table 4. In comparison to the PI film without β-CD, the introduction of excessive cyclodextrin adversely impacts the polymerization of PAA, resulting in a lower molecular weight, increased formation of ring chains or short chains, and a plasticizing effect akin to small molecules. This effect reduces the entanglement of molecular chains, consequently leading to increased elongation at break. Although the equimolar ratio of diamines (50%:50%) may significantly disrupt molecular regularity, it may compromise both the modulus and the strength.

### 3.4. Thermal Properties of β-Cyclodextrin PI Composite Films

The thermogravimetric curve graph of β-CD/PI shows that the polyrotane system started to decrease only after surpassing 300 °C, as shown in Figure 7b, indicating the complete imidization of PI film with no residual solvent. In addition, the initial decomposition temperature of the polyrotane system reaches 330 °C. Since the decomposition temperature of cyclodextrin is approximately 250 °C, this observation supports the absence of free β-CD, instead, forming a round-ended structure nested on the PI molecular chain. The trifluoromethyl structure in HFBAPP and the chair imide ring contribute substantial spatial hindrance, anchoring β-CD firmly onto the main chain of polyimide.

Via thermogravimetric analysis, the mass fraction of cyclodextrin in the system is calculated. It can be seen from Figure 6b that the β-CD content is 11.49% in PI-CD2. With the increase in the initial cyclodextrin addition, the mass fraction of cyclodextrin in the system decreases. Under the same condition, cyclodextrin cannot be effectively dispersed, reducing the probability of cyclodextrin inclusion of diamine monomer. A large number of free cyclodextrin is eluted and removed in the subsequent process.

For further investigation, the H″PMDA-HFBAPP/ODA-PI copolymer system showed remarkable thermal stability, and it was observed that the initial decomposition temperature of the system increased with the increase in the HFBAPP addition ratio. Among these samples, the PI-10 variant demonstrated superior heat resistance, achieving a T_d5%_ of 532.31 °C, with a residual weight rate of 43.77% at 800 °C, as shown in Figure 7a. This heightened thermal stability can be attributed to the abundant trifluoromethyl and aromatic groups present in the molecular structure of HFBAPP.

Under the large steric hindrance, the molecular chain is difficult to rotate and move, and the introduction of HFBAPP leads to an increase in molecular weight and crosslinking density, contributing to the enhancement of thermal stability. Table 5 shows that the improvement of T_d5%_ in PI-10 compared to PI-7 is not as high as that in PI-5 to PI-3. With the increase in fluorine content, the improvement of heat resistance slows down, which may be due to the decrease in crosslinking density caused by excessive HFBAPP and the higher ether bond content in HFBAPP relative to ODA, which enhances the mobility of polymer chains. However, due to the increase in the content of the rigid structure of the benzene ring in the polymer, the residual weight rate gradually increases at 800 °C.

As shown in Figure 7d and Table 5, the introduction of HFBAPP can improve the glass transition temperature of polyimides. Due to the structural presence of ether bonds and trifluoromethyl moieties, with the increase in HFBAPP content, the glass transition temperature of the system also increases.

According to Figure 7b, it is observed that PI-CD2, characterized by the highest β-CD content, exhibits a transmittance of only 73.7% at 450 nm and 84.8% at 800 nm. Although the addition of β-CD has some side effects on the optical transmittance of the copolymer system, the optical transmittance of the polyrotane system is still significantly better than that of the traditional Kapton PI.

### 3.5. Optical Properties and Dissolution Performance

After applying the polyimide solutions onto a glass plate, colorless transparent films were obtained following the curing process.

The UV-visible transmission spectra of these cast films indicated that the optical transmittances were greater than 80% at 350 nm and 85% at 400 nm (45 μm thickness), making them colorless in practice, and the digital photos of polyimide films and β-CD/PI films (Appendix A) were colorless.

PI film coloring issues were eliminated by interfering with CT interactions using aliphatic (usually alicyclic) monomers in diamine, tetracarboxylic dianhydride, or both [22,24]. The transmittance of H″PMDA-HFBAPP/ODA-PI was evaluated based on its UV cutoff wavelength and transmittance at 450 nm, as shown in Figure 8. The lower UV cutoff wavelength and higher transmittance values at 450 nm indicated a better optical transmittance performance. In Figure 8, the UV–visible transmission spectrumβ-CD/PI reveals a certain reduction in optical transparency compared to the pure PI system, and the degree of decrease is similar to that of the pure PI system, as it is negatively correlated with the addition of cyclodextrin.

The β-CD-PI composite films, exhibiting polyrotaxane structures, demonstrate exceptional solubility not only in common high boiling polar aprotic solvent like DMAC and DMSO but also in common low boiling polar solvents such as chloroform (CHCl_3_) and dichloromethane (CH_2_Cl_2_). Compared with traditional all-aromatic polyimides, the solubility of PI-0 synthesized in the previous chapter has been greatly improved in Table 6. The introduction of an alicyclic ring with a large space volume, trifluoromethyl groups, and flexible ether bond reduces the stacking density between PI molecular chains and improves the flexibility and fluidity of the PI molecule. The introduction of cyclodextrin further enlarges the volume of the PI molecular chain, diminishing the interaction between molecular chains and enhancing the flexibility and fluidity of the molecular chain segments. The significant improvement in solubility and processability of PI films allows for the substitution of traditional highly toxic organic solvents, commonly used in polyimide synthesis, with solvents of low toxicity and boiling point. This substitution not only reduces the complexity of PI industrial production but also contributes to environmental friendliness.

### 3.6. Dielectric Properties of β-Cyclodextrin PI Composite Films

Dielectric properties are inherently influenced by the chemical structures of polymers, and PIs exhibit varying dielectric constants based on their molecular compositions. In our study, the dielectric performance of polyimide was meticulously examined in the frequency range of 40–10 MHz. The dielectric permittivity exhibited a discernible trend, decreasing from 2.74 for the pristine polyimide to 2.65 for the composite polyimide containing β-CD, as the β-CD content increased in Figure 9.

The robust electron absorption capability inherent to F atoms renders the polarization of C-F bonds challenging. This pronounced repulsion between F atoms serves to impede the stacking morphology between chain segments, fostering the creation of more expansive, loosely structured regions within the material, increasing the free volume inside the material, and thus leading to a decrease in the dielectric constant. This strategic manipulation of fluorine content presents an effective means to systematically diminish the dielectric constant, underscoring the intricate interplay between molecular composition and dielectric properties [32].

A discernible trend in dielectric constants is noted, exhibiting a decrease followed by an increase as the β-CD doping amounts vary. The augmentation of β-CD content results in a uniform integration of cyclodextrin molecules within the molecular chain, thereby augmenting the free volume of the PI matrix and concomitantly diminishing the dielectric constant. However, as the content of β-CD continues to increase, the formation of the nested structure becomes less uniform.

In addition, the lateral length serves as an indicator of the molar volume of the material. A larger lateral length corresponds to a more distorted molecule, leading to increased gaps between molecular chains and enhanced free volume within the structure. Additionally, the dispersion of the electron cloud between molecules becomes more pronounced, impeding alterations in molecular dipole in response to an electric field and consequently contributing to a reduction in dielectric constant.

The dielectric data of the polyimides are shown in Table 7. Because of the strong electron absorption and large free volume, the β-CD and -CF_3_ group is considered to be an effective way to reduce the dielectric constant. It can be seen that all prepared β-CD/PI films showed dielectric constants below 3.0, meeting the need of the high-frequency communication substrates.

According to the Clausius Mossotti equation in Equation (2), the chain polarizability (P) and free volume (V) of the polymer are the main factors affecting its dielectric constant k [33]. According to the Equation (2), introduced β-cyclodextrin effectively expands the free volume of PI molecular chain on the basis of the synthesis co-polyimide, which reduced the molar polarizability per unit volume effectively and the dielectric constant of PI films. The change in water contact angle corresponds to the increase in HFBAPP content (Appendix A).

## 4. Conclusions

This study presented an innovative design strategy aimed at effectively reducing the dielectric constant of PI films. The strategy revolves around using β-CD to augment the free volume, particularly the rotational structure inherent in the β-CD/diamine (ODA, HFBAPP) pendant group units, thereby enhancing the overall free volume within the bulk. The successful implementation of this novel approach has yielded a PI (β-CD/co-PI) with outstanding low dielectric properties. The dielectric constant (k) and dielectric loss (tanδ) of the β-CD/PI film are impressively low, measuring 2.486 and 0.00186, respectively, at 10 MHz. Notably, the PI films maintain exceptional thermal and mechanical characteristics while exhibiting a low dielectric profile across a broad frequency range. This strategic design not only effectively lowers the dielectric constant (k) but also preserves the overall advantageous properties of polyimides. We believe that this innovative approach holds promise for extension to other high-performance polymer systems. The commendable dielectric, mechanical, and thermal attributes position PI/β-CD composite films as promising contenders for flexible thin-film all-organic capacitors, even in high-temperature environments.

## Figures and Tables

**Figure 1 polymers-16-00901-f001:**
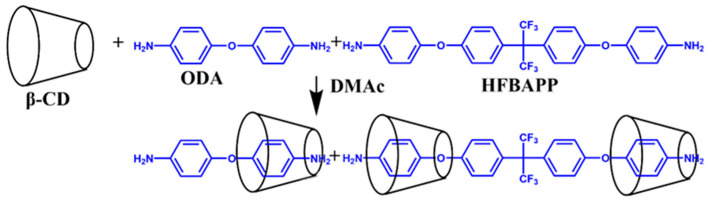
Synthesis routes of the complexation of diamine monomer with β-cyclodextrin.

**Figure 2 polymers-16-00901-f002:**
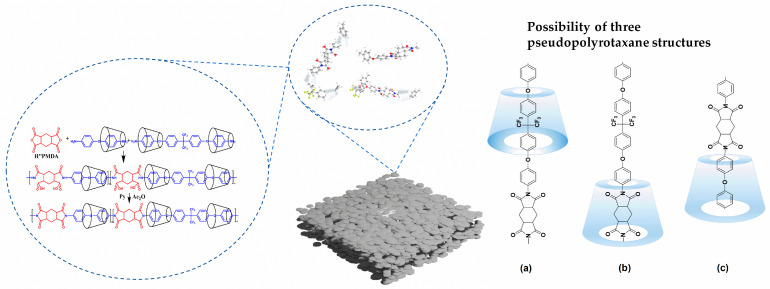
The polymerization of diamines, dianhydrides, and β-CDs and rotaxane structures of (**a**) HFBAPP section with β-CD, (**b**) H″PMDA section with β-CD, and (**c**) ODA section with β-CD [19].

**Figure 3 polymers-16-00901-f003:**
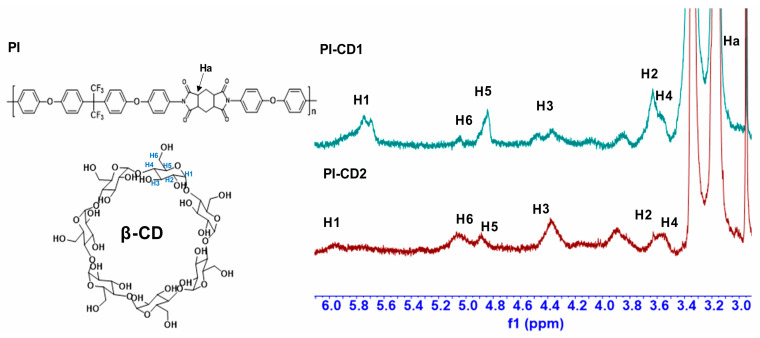
^1^H NMR spectrum of PI with different cyclodextrin content. H1–H6 is the protons from β-CD, Ha is the proton from PI.

**Figure 4 polymers-16-00901-f004:**
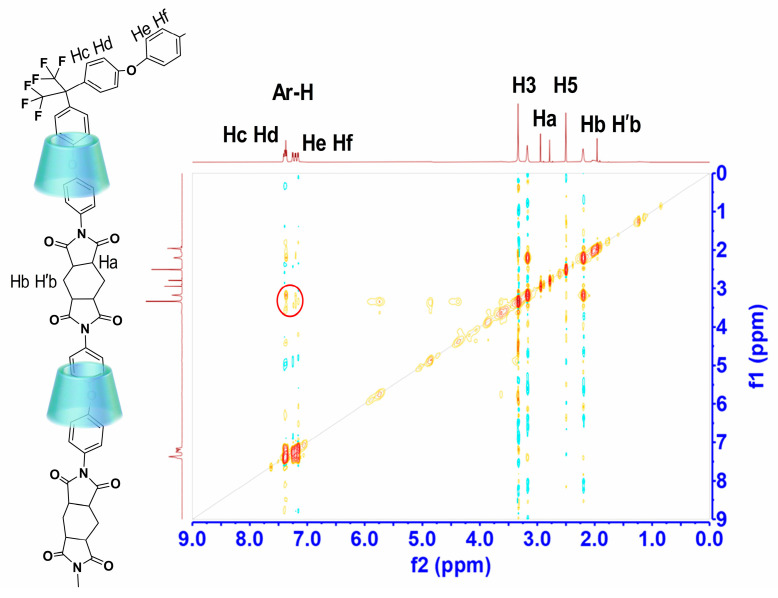
Two-dimensional ^1^H NMR spectrum of β-CD-H″PMDA-HFBAPP/ODA-PI. The cross peak in red circle showing the interaction of the Ar-H with the β-CD inner protons.

**Figure 5 polymers-16-00901-f005:**
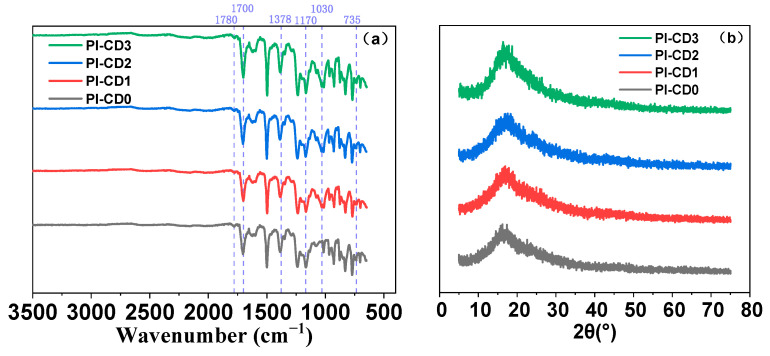
(**a**) FTIR and (**b**) XRD spectra of PI and β-CD/PI composite films.

**Figure 6 polymers-16-00901-f006:**
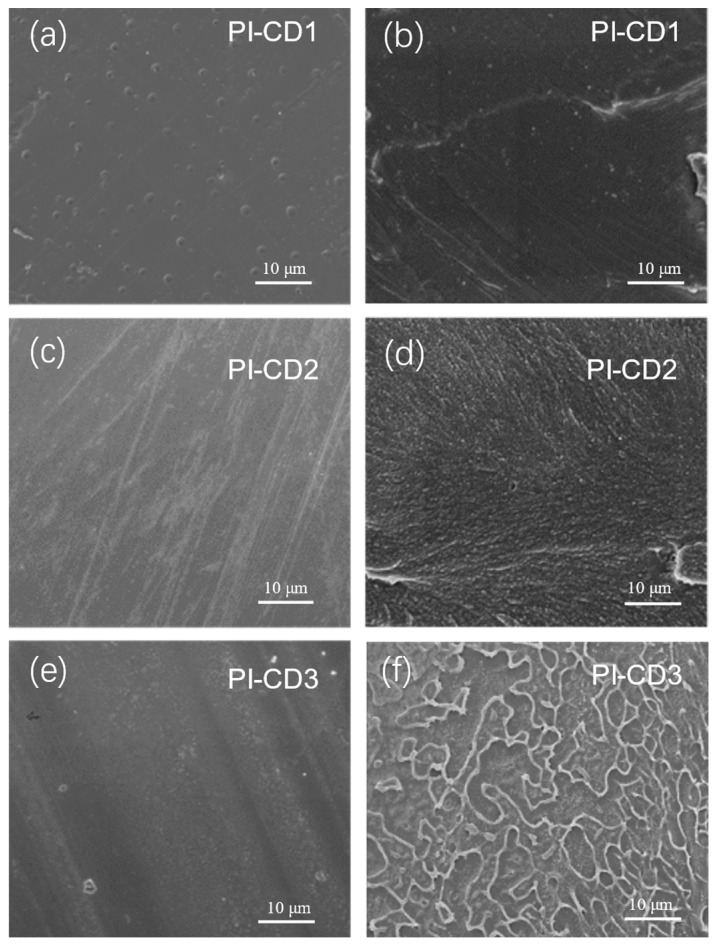
Surface and cross-sectional SEM images of β-CD/PI composite films with different β-CD content, (**a**): surface-sectional SEM images (20 molar % content of β-CD in PI matrix), (**b**): cross-sectional SEM images (20 molar % content of β-CD in PI matrix), (**c**): surface-sectional SEM images (30 molar % content of β-CD in PI matrix), (**d**): cross-sectional SEM images (30 molar % content of β-CD in PI matrix), (**e**): surface-sectional SEM images (50 molar % content of β-CD in PI matrix), (**f**): cross-sectional SEM images (50 molar % content of β-CD in PI matrix).

**Figure 7 polymers-16-00901-f007:**
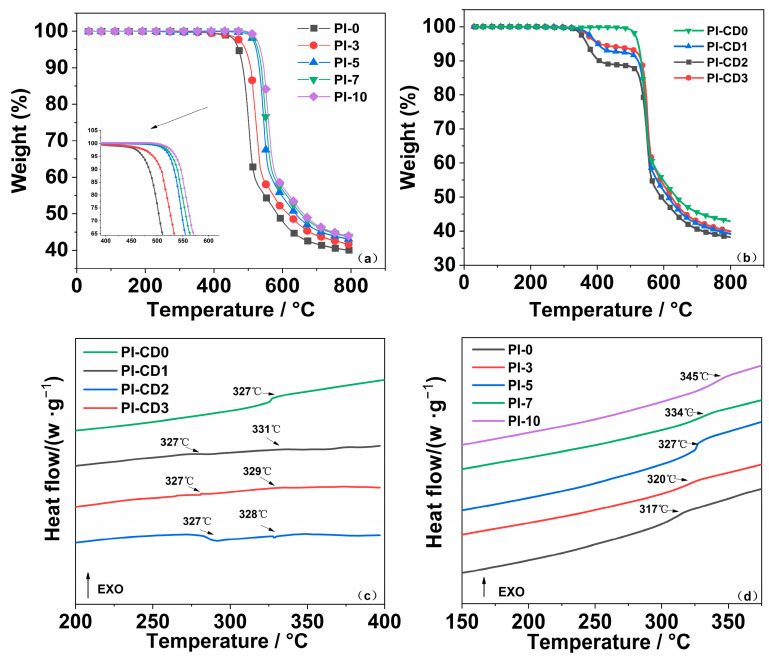
(**a**): TGA curves of PI films, (**b**): TGA curves of β-CD/PI composite films, (**c**): DSC curves of β-CD/PI films, (**d**): DSC curves of PI film.

**Figure 8 polymers-16-00901-f008:**
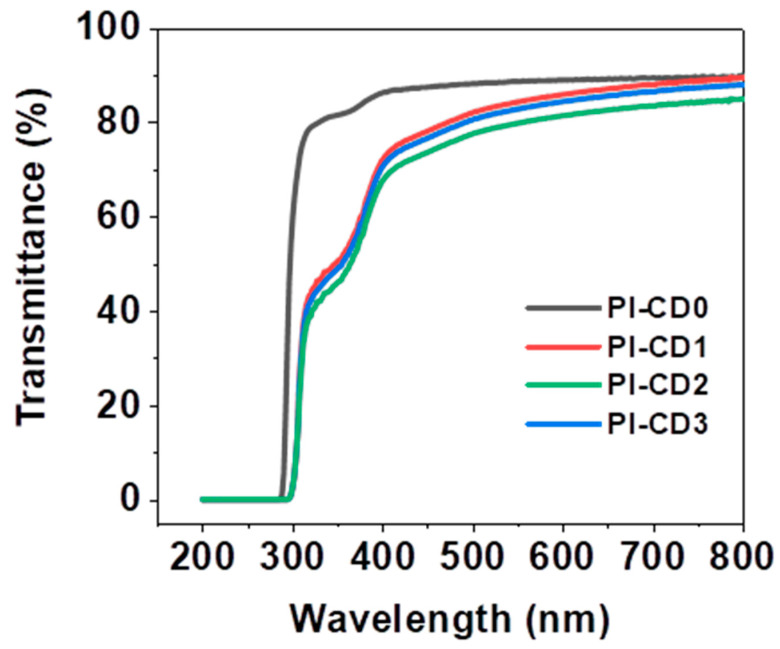
UV–Vis spectra of β-CD/PI composite films.

**Figure 9 polymers-16-00901-f009:**
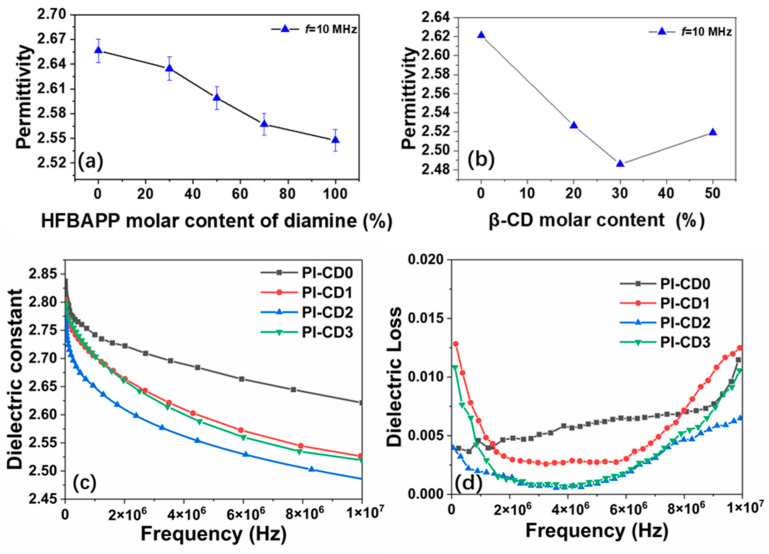
Effect of (**a**) HFBAPP monomer content and (**b**): β-CD content on dielectric constant property of PI or PI composite films; (**c**): dielectric constant and (**d**): dielectric loss of PI or PI composite films.

**Table 1 polymers-16-00901-t001:** Molar ratio of the H″PMDA/ODA/HFBAPP section.

System Name	n(diamine) (mmol)	n(ODA) (mmol)	n(HFBAPP) (mmol)	n(β-CD) (mmol)	n(H″PMDA) (mmol)	n(HFBAPP)	n(βCD)/n(ODA)
PI-0	2.504	2.504	0	——	2.504	0	——
PI-3	2.504	0.7512	1.7528	——	2.504	0	0.5
PI-5	2.504	1.252	1.252	——	2.504	0	1
PI-7	2.504	1.7528	0.7512	——	2.504	0	2
PI-10	2.504	0	2.504	——	2.504	0	2
PI-CD0	5.008	2.504	2.504	——	5.008	——	——
PI-CD1	5.008	2.504	2.504	1.252	5.008	0.5	0.5
PI-CD2	5.008	2.504	2.504	2.504	5.008	1	1
PI-CD3	5.008	2.504	2.504	5.008	5.008	2	2

——: No data here.

**Table 2 polymers-16-00901-t002:** The properties of polyimides synthesized by other researchers.

Dianhydride	Diamine	T_400_(%)	λ_Cut_(nm)	ε	E(GPa)	T^5^_d_(N_2_)	T_g_(°C)	Ref.
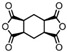	4.4′-ODA	84	293	——	2.4	442	333	[21]
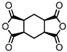	TFMB	85	292	——	2.7	474	370	[21]
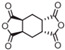	3.4′-ODA	87.8	306	2.86	1.69	——	253	[22]
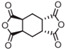	4.4′-ODA	92	306	2.82	2.31	——	306	[22]
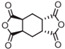	TFMB	89.5	291	2.65	2.23	477	291	[22]
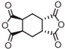	HFBAPP	7	296	2.75	1.59	475	277	[22]
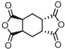	4.4′-ODA/HFBAPP	85	286	2.65	2.47	523	327	This work
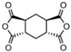	BMD	76	335	——	2.8	465	421	[23]
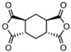	BMD	80	330	——	2.8	465	409	[23]
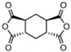	4,4′-ODA	76	294	2.88	2.42	453	294	[24]
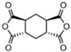	HFBAPP	87.7	286	2.72	1.56	502	287	[24]
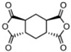	TFMB	90.3	292	2.64	——	455	344	[24]
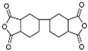	4,4′-ODA	80	298	2.81	2.1	468	256	[25]
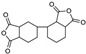	4,4′-ODA	78	310	2.77	2.5	446	321	[25]
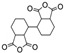	4,4′-ODA	83	299	2.81	2	438	297	[25]

——: No data here.

**Table 3 polymers-16-00901-t003:** Molecular weight data of β-CD/PI.

Sample	[η]/dL·g^−1^	Mn	Mw	PDI
PI-CD0	2.76	2.9 × 10^5^	4.0 × 10^5^	1.39
PI-CD1	1.59	1.5 × 10^5^	2.1 × 10^5^	1.42
PI-CD2	1.55	1.4 × 10^5^	1.9 × 10^5^	1.42
PI-CD3	1.38	1.1 × 10^5^	1.9 × 10^5^	1.77

**Table 4 polymers-16-00901-t004:** Mechanical properties of β-CD/H″PMDA-HFBAPP/ODA-PI composite films.

Sample	Tensile Strength (MPa)	Tensile Modulus (GPa)	Elongation at Break (%)	Thickness (μm)
PI-CD0	145	2.47	6.3	120
PI-CD1	124	2.23	6.5	130
PI-CD2	122	2.20	6.8	115
PI-CD3	98	1.87	7.4	140

**Table 5 polymers-16-00901-t005:** Thermal properties of β-CD-PI composite films.

Sample	T_d5%_/°C	T_d10%_/°C	R_800c_/%	Tg/°C	Wt%(β-CD)
PI-0	472	484	39.93	317	0
PI-3	492	507	41.48	320	0
PI-5	523	532	42.89	327	0
PI-7	528	539	43.63	334	0
PI-10	532	546	43.77	345	0
PI-CD0	524	533	42.89	327	0
PI-CD1	401	523	39.19	331	7.63
PI-CD2	370	407	38.18	328	11.49
PI-CD3	407	533	39.92	329	6.33

**Table 6 polymers-16-00901-t006:** Solubility of H″PMDA-HFBAPP/ODA-PI and β-CD/PI.

Sample	DMAc	DMF	DMSO	NMP	CH_2_Cl_2_	CHCl_3_	THF
PI-0	+−	+−	+−	+−	−−	−−	−−
PI-3	++	++	++	++	−−	−−	−−
PI-5	++	++	++	++	−−	−−	−−
PI-7	++	++	++	++	+−	+−	+−
PI-10	++	++	++	++	+−	+−	+−
PI-CD0	++	++	++	++	+−	+−	−−
PI-CD1	++	++	++	++	++	++	+−
PI-CD2	++	++	++	++	++	++	++
PI-CD3	++	++	++	++	++	++	+−

++: Soluble at R.T.; +−: soluble after heating; −−: partially soluble after heating.

**Table 7 polymers-16-00901-t007:** Dielectric properties of β-CD/PI composite films.

Sample	β-CD ^a^ (mol%)	K (10 MHz)	tanθ (10 MHz)	F ^b^ (wt%)
PI-CD0	0	2.6211	0.00395	3.198
PI-CD1	20	2.5264	0.00482	4.785
PI-CD2	30	2.486	0.00186	5.989
PI-CD3	50	2.5192	0.00291	7.457

^a^: Mole fraction of β-Cyclodextrin added in the system, ^b^: mass fraction of fluorine added to the system.

## Data Availability

Data are contained within the article.

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
