# Peer review of "Polyimide Films Based on β-Cyclodextrin Polyrotaxane with Low Dielectric and Excellent Comprehensive Performance"

_polymers, 2024, doi:10.3390/polym16070901_

Round 1
Reviewer 1 Report
Comments and Suggestions for Authors
Point 1:- As you mentioned β-CD for polyimide films, instead of β-CD can we able to take other cyclodextrin like γ-Cyclodextrin and α-Cyclodextrin. Is it feasible or only beta cyclodextrin has such properties
Point 2:- Mention Polyimide films (thickness and other properties)
Point 3:- CD ratio 1:1 inclusion complex with the diamine monomer; Did other ration was taken in consideration
Point 4:- in figure 5, see that legends should be just below images.
Point 5:- Somo typological errors such as in line 232, -450 etc...
Author Response
Response to Reviewer #1:
- As you mentioned β-CD for polyimide films, instead of β-CD can we able to take other cyclodextrin like γ-Cyclodextrin and α-Cyclodextrin. Is it feasible or only beta cyclodextrin has such properties.
Response: Thank you so much for valuable suggestion. We concur with this observation. As such, the inner diameter of α-cyclodextrin is 0.5nm, and that of γ-cyclodextrin is 0.8nm. The chain spacing of the polyimide film was determined from the angular values of the diffraction peaks of the X-ray diffraction spectra as 5.1013-5.2688Å. Consequently, it is unlikely that α-cyclodextrin could form a nested structure with it, and γ-cyclodextrin would detach from it during subsequent washing due to the large diameter of the inner cavity. I have included a discussion of the possibility of nested formation of the three cyclodextrins on page 7, first paragraph, line 291. Thank you again for your valuable suggestion to make our work more complete. (line 291).
- Mention Polyimide films (thickness and other properties).
Response: Thank you for your suggestion. Therefore, we have added a discussion of the properties of polyimides to ensure the completeness of the results. We have incorporated the polyimide film thickness information into Table 3, as indicated on page 13, line 446. And the water contact angle results of fluorinated aliphatic copolymer polyimide films were added in the Supporting Information (Figure S6).
- CD ratio 1:1 inclusion complex with the diamine monomer; Did other ration was taken in consideration.
Response: Thank you for your suggestion. It is important to consider the possibility of non-1:1 nested ratio in our analysis. This allows for a more complete and accurate assessment of the data. It's important to acknowledge the potential for a 1:2 ratio of inclusion and non-inclusion of the diamine monomer when considering its steric configuration. Due to the inner diameter of cyclodextrin and molecular size of the diamine monomer (ODA or HFBAPP), it is assumed that cyclodextrins encapsulate the diamine monomer in a 1:1 ratio.
- in figure 5, see that legends should be just below images..
Response: Thank you for your suggestion. We have accordingly revised the legend position to emphasize this point, ensuring clear communication of the assumptions and limitations in our work. We have updated Figure 5 on page 10, line 367 with a new legend. We are very sorry for our incorrect writing (line 367).
- Somo typological errors such as in line 232, -450 etc.
Response: Considering the Reviewer’s suggestion, We have revised to emphasize this point accordingly. (line 232-450).

Reviewer 2 Report
Comments and Suggestions for Authors
In this paper, a two-step method for the integration of β-cyclodextrin into semi-aromatic fluorine-containing polyimide ternary systems was used. By introducing trifluoromethyl groups to reduce the dielectric constant was further reduced to 2.55 at 10 MHz. Meanwhile, the film exhibited remarkable thermal stability (glass transition temperature of more than 300 °C) and high coefficient of thermal expansion. The material also demonstrated outstanding mechanical properties: strength of 122 MPa and elastic modulus of 2.2 GPa, as well as high optical transparency (transmittance reaches 89% at a wavelength of 450 nm).
The work is complete with a full set of methods and proper analyses of the results obtained.
There are observations:
1. For all devices, a clarification is required related to the specification of the manufacturer and the country of manufacture.
2. Figures 5 and 7 require additional clarification.
3. A table of comparison with the works of other researchers on improving the properties of polyamides should be given.
Comments on the Quality of English Language
Minor editing of English language required
Author Response
Response to Reviewer #2:
- In this paper, a two-step method for the integration of β-cyclodextrin into semi-aromatic fluorine-containing polyimide ternary systems was used. By introducing trifluoromethyl groups to reduce the dielectric constant was further reduced to 2.55 at 10 MHz. Meanwhile, the film exhibited remarkable thermal stability (glass transition temperature of more than 300 °C) and high coefficient of thermal expansion. The material also demonstrated outstanding mechanical properties: strength of 122 MPa and elastic modulus of 2.2 GPa, as well as high optical transparency (transmittance reaches 89% at a wavelength of 450 nm). The work is complete with a full set of methods and proper analyses of the results obtained.
Response: Thanks for the excellent summary of our work.
- For all devices, a clarification is required related to the specification of the manufacturer and the country of manufacture.
Response: Agree. We have modified clarification of all devices to emphasize this point accordingly. Lines 212, 214, 216, 219, 225, 229 on page 5 and lines 231, 237, 243, 246 on page 6 add country-specific information for all qualitative tools used. (line 212-246).
- Figures 5 and 7 require additional clarification.
Response: Thank you for your suggestion. Consequently, I made adjustments to the layout of the legend to enhance clarity. We have changed the information in the title of the chart in lines 367 and 368 on page 10 to correspond to the legend in figure 5; And on page 13, lines 456, 457 correspond to the legend in figure 7, and on page 17, lines 565, 567 correspond to Figure 9. The figures has been corrected to clarify the information in the figure. (line 367-368, 565-567).
- A table of comparison with the works of other researchers on improving the properties of polyamides should be given.
Response: Thank you for your suggestion, it is important to study the performance of similarity for PI films. A cross-reference to the work done by other researchers in improving the properties of polyamides is also added to the manuscript on page 7 (line 297).

Round 2
Reviewer 2 Report
Comments and Suggestions for Authors
Accept in present form